# Synthesis, Structure and Photoluminescence Properties of Cd and Cd-Ln Pentafluorobenzoates with 2,2′:6′,2′-Terpyridine Derivatives

**Maxim A. Shmelev** [1], **Julia K. Voronina** [1], **Maxim A. Evtyukhin** [2], **Fedor M. Dolgushin** [1], **Evgenia A. Varaksina** [3], **Ilya V. Taydakov** [3], **Aleksey A. Sidorov** [1], **Igor L. Eremenko** [1] and **Mikhail A. Kiskin** [1,*]

1   N. S. Kurnakov Institute of General and Inorganic Chemistry, Russian Academy of Sciences,
    31 Leninsky prosp., 119991 Moscow, Russia
2   Department of Chemistry and Technology of Rare Elements, M.V. Lomonosov Institute of Fine Chemical
    Technologies, MIREA—Russian Technological University, 78 Vernadsky Avenue, 119454 Moscow, Russia
3   P. N. Lebedev Physical Institute, Russian Academy of Sciences, 53 Leninsky prosp., 119991 Moscow, Russia
*   Correspondence: mkiskin@igic.ras.ru

**Abstract:** Six new complexes [Cd(tpy)(pfb)$_2$] (**1**, tpy = 2,2′:6′,2′′-terpyridine), [Ln$_2$Cd$_2$(tpy)$_2$(pfb)$_{10}$] (Ln = Eu (**2$_{Eu}$**), Tb (**2$_{Tb}$**)), [Ln$_2$Cd$_2$(tbtpy)$_2$(pfb)$_{10}$]·2MeCN (Ln = Eu (**3$_{Eu}$**), Tb (**3$_{Tb}$**), tbtpy = 4,4′,4′′-tri-*tert*-butyl-2,2′:6′,2′′-terpyridine), [Eu$_2$Cd$_2$(tppz)(pfb)$_{10}$]$n$ (**4**, tppz = 2,3,5,6-tetra-(pyridin-2-yl)pyrazine) based on pentafluorobenzoic acid (Hpfb) have been prepared and investigated. The effect of tridentate ligands on geometry heterometallic scaffolds synthesized complexes is discussed. The supramolecular crystal structures of the new compounds are stabilized by π-π, C-F⋯π, C-H⋯O, C-H...F, F....F interactions. Non-covalent interactions have been studied using Hirschfeld surface analysis. The obtained compounds were characterized by single-crystal and powder X-ray diffraction, luminescence spectroscopy, IR spectroscopy, CHN analysis. Complexes **2$_{Ln}$** and **3$_{Ln}$** exhibit metal-centered photoluminescence, but the presence of ligand luminescence bands indicates incomplete energy transfer from the *d*-block to the lanthanide ion.

**Keywords:** Cadmium(II); Lanthanide(III); heterometallic complexes; coordination polymers; pentafluorobenzoic acid; X-ray study; non-covalent interactions; Hirshfeld surface analysis; photoluminescence

## 1. Introduction

The actual direction of research in the field of chemistry and materials science is the design and study of compounds with desired properties. It is known that compounds of lanthanides have unique physical and chemical properties and can be used as functional materials in various fields: lasers, lighting systems, electroluminescent devices and diodes, sensors, catalysis, biomedical imaging, magnetic refrigerators, and much more [1–5]. Moreover, 4*f* complexes attract attention as polyfunctional compounds that exhibit both magnetic and luminescent properties [6–8]. The additional introduction of d-metal ions and the preparation of heterometallic {*d*$^{10}$-Ln} complexes significantly expand the possibilities of practical application of the obtained compounds, since the introduction of a d-block with organic antenna molecules into the REE coordination environment contributes to a change in the energy value of the triplet level of the organic ligand, which affects the efficiency of energy transfer to the Ln ion [9,10].

The problems of synthesizing a predetermined structure are associated with both molecular design and the use of various types of non-covalent interactions (hydrogen, halogen bonds, stacking interactions and others) to control the geometry and crystal packing of complexes [11–16]. We drew attention to the known systems of organic compounds that combine fluorinated and non-fluorinated aromatic fragments [17–20]. These systems are characterized by dense stack crystal packing of aromatic cycles and their convergence to a distance of 3.4–3.6 Å due to strong non-covalent interactions.

In the case of REE complexes with fluorinated ligands, one can expect an improvement in the photoluminescence characteristics of the synthesized compound, since the use of pentafluorobenzoic acid leads to a decrease in luminescence quenching due to the absence of high-frequency C–H vibrations [21,22] and the resulting strong stacking interactions can contribute to an increase in luminescence quantum yields due to additional stabilization of the crystalline packaging [23–25].

Our recent studies show that a number of Cd, Ln pentafluorobenzoic complexes and heterometallic Ln-Cd pentafluorobenzoic compounds are capable of forming unusual molecular and polymeric structures [25–27]. The unusual structure of these compounds is largely determined by the structure-forming effects of non-covalent interactions. Moreover, these interactions lead to forming homo- and heterometal coordination polymers in some cases; composition once was typical for molecular complexes, and in other cases to destruction of polynuclear structures, which with other monocarboxylic acid anions show high stability. These interactions led in some cases to the formation of homo- and heterometallic coordination polymers, the composition of which was typical for molecular complexes, while in others they led to the destruction of polynuclear structures, which, in the case of other anions of monocarboxylic acids exhibit high stability. By establishing combinations of pentafluorobenzoic anions and other aromatic ligands in the structure, it is possible to obtain heteroanionic molecular complexes or coordination polymers with stacked packing of aromatic fragments. It should be noted that in the case of both heteroanionic complexes and the case of coordination polymers, an increase in luminescence quantum yields was shown. Accordingly, the synthesis and study of compounds in which pentafluorobenzoic anions combine with various aromatic ligands are of interest both for studying the formation of new and unusual structures, and for obtaining compounds that are promising as photoluminescent materials.

The study of pentafluorobenzoic complexes with tridentate pincer N-donor ligands is a logical continuation of our work, in which numerous pentafluorobenzoic complexes with monodentate and bidentate-chelate N-donor ligands were synthesized. In this paper, we show new examples of the use of 2,2′:6′,2′′-terpyridine, 4,4′,4′′-tri-tert-butyl-2,2′:6′,2′′-terpyridine, 2,3,5,6-tetra-(pyridin-2-yl)pyrazine for the synthesis of Cd and Ln-Cd pentafluorobenzoate complexes.

## 2. Experimental

### 2.1. Materials and Methods

All synthetic work was performed in air. Commercially available reagents were used as received, 2,2′:6′,2′′-terpyridine (tpy, 98%, Sigma-Aldrich Chemie, Steinheim, Germany), 4,4′,4′′-tri-*tert*-butyl-2,2′:6′,2′′-terpyridine (tbtpy, 95%, Sigma-Aldrich Chemie, Steinheim, Germany), 2,3,5,6-tetra-(pyridin-2-yl)pyrazine (tppz, 97%, Sigma-Aldrich Chemie, Steinheim, Germany), EtOH (96%, Ferein, Elektrogorsk, Moscow region, Russia), MeCN ($\geq$99.5%, "Chimmed, Moscow, Russia"). The initial substances [Eu$_2$Cd$_2$(MeCN)$_4$(pfb)$_{10}$], [Tb$_2$Cd$_2$(MeCN)$_6$(pfb)$_{10}$]·4MeCN and [{Cd(H$_2$O)$_4$(pfb)}$_n$·$n$(pfb)] were synthesized according to the literature method [26,28].

Elemental analysis was carried out on an EA1108 Carlo Erba automatic CHNS-analyzer (EuroVector, Pavia PV, Italy). IR spectra of the compounds were recorded on a Perkin Elmer Spectrum 65 spectrophotometer (Perkin Elmer, Waltham, MA, USA) equipped with a Quest ATR Accessory (BR5 3FQ, Specac, Orpington, UK) by the attenuated total reflectance (ATR) in the range 400–4000 cm$^{-1}$. Luminescent spectra were measured with a Perkin Elmer LS-55 spectrofluorimeter (Perkin Elmer, Waltham, MA, USA).

### 2.2. Synthesis of the Compounds

2.2.1. [Cd(tpy)(pfb)$_2$] (1)

Tpy (0.039 g, 0.166 mmol) was added to the solution [{Cd(H$_2$O)$_4$(pfb)}$_n$·$n$(pfb)] (0.100 g, 0.166 mmol) in 5 mL of MeCN. The reaction mixture was stirred for 30 min and kept in the sealed vial at 70 °C. Colourless crystals suitable for X-ray analysis were obtained after

2 days. The crystals were collected by decantation and washed with cold MeCN ($t$ = ~5 °C). The yield of **1** was 0.103 g (81%) based on [{Cd(H$_2$O)$_4$(pfb)}$_n$·$n$(pfb)]. Calculated for C$_{29}$H$_{11}$O$_4$N$_3$F$_{10}$Cd (%): C, 45.4; H, 1.4; N, 5.5. Found (%): C, 45.3; H, 1.2; N, 5.7. IR ($\nu$, cm$^{-1}$): 3484 w, 3074 w, 1595 s, 1501 s, 1481 s, 1316 w, 1252 m, 1193 w, 1165 w, 1099 m, 989 s, 927 m, 828 w, 762 s, 652 w, 582 w, 508 m, 470 m.

### 2.2.2. [Eu$_2$Cd$_2$(tpy)$_2$(pfb)$_{10}$] (2$_{Eu}$)

Tpy (0.013 g, 0.054 mmol) was added the solution of [Eu$_2$Cd$_2$(MeCN)$_4$(pfb)$_{10}$] (0.150 g, 0.054 mmol) in 10 mL of MeCN. The reaction mixture was stirred for 20 min and kept in the sealed vial at 70 °C. Colourless crystals suitable for X-ray analysis were obtained after 12 days. The crystals were collected by decantation and washed with cold MeCN ($t$ = ~5 °C). The yield of **2$_{Eu}$** was 0.059 g (36%) based on [Eu$_2$Cd$_2$(MeCN)$_4$(pfb)$_{10}$]. Calculated for C$_{100}$H$_{22}$O$_{20}$N$_6$F$_{50}$Eu$_2$Cd$_2$ (%): C, 38.6; H, 0.7; N, 2.7. Found (%): C, 38.9; H, 0.9; N, 2.6. IR ($\nu$, cm$^{-1}$): 3675 w, 2901 w, 1651 m, 1618 m, 1589 s, 1523 m, 1491 s, 1449 m, 1404 s, 1313 w, 1294 m, 1162 m, 1108 s, 993 s, 937 m, 826 m, 769 m, 751 m, 741 s, 696 s, 652 m, 634 m, 583 m, 507 m.

### 2.2.3. [Tb$_2$Cd$_2$(tpy)$_2$(pfb)$_{10}$] (2$_{Tb}$)

Tpy (0.012 g, 0.049 mmol) was added to the solution of [Tb$_2$Cd$_2$(MeCN)$_6$(pfb)$_{10}$]·4MeCN (0.150 g, 0.049 mmol) in 10 mL of MeCN. The reaction mixture was stirred for 20 min and kept in the sealed vial at 70 °C. The white precipitate formed after 2 days was separated from the mother liquor by decantation and washed with cold MeCN ($t$ = ~5 °C). The yield of **2$_{Tb}$** was 0.061 g (38%) based on [Tb$_2$Cd$_2$(MeCN)$_6$(pfb)$_{10}$]·4MeCN. Calculated for C$_{100}$H$_{22}$O$_{20}$N$_6$F$_{50}$Tb$_2$Cd$_2$ (%): C, 38.5; H, 0.7; N, 2.7. Found (%): C, 38.6; H, 0.5; N, 2.5. IR ($\nu$, cm$^{-1}$): 2901 w, 1650 m, 1618 m, 1590 s, 1523 m, 1491 m, 1449 m, 1403 s, 1313 w, 1294 m, 1162 m, 1109 s, 993 s, 935 w, 826 m, 769 m, 751 w, 741 s, 696 s, 651 m, 635 m, 583 m, 506 m.

### 2.2.4. [Eu$_2$Cd$_2$(tbtpy)$_2$(pfb)$_{10}$]×2MeCN (3$_{Eu}$)

The **3$_{Eu}$** was obtained according to the synthetic method similar to **2$_{Eu}$**, using tbtpy (0.031 g, 0.108 mmol) instead tpy. The yield of **3$_{Eu}$** was 0.101 g (54%) based on [Eu$_2$Cd$_2$(MeCN)$_4$(pfb)$_{10}$]. Calculated for C$_{128}$H$_{76}$O$_{20}$N$_8$F$_{50}$Eu$_2$Cd$_2$ (%): C, 43.6; H, 2.2; N, 3.2. Found (%): C, 43.5; H, 2.1; N, 3.3. IR ($\nu$, cm$^{-1}$): 2969 w, 1929 w, 1737 m, 1679 m, 1618 s, 1593 s, 1493 s, 1295 m, 1249 m, 1202 w, 1108 m, 993 s, 932 m, 882 w, 828 m, 761 s, 741 s, 695 m, 615 w, 583 w, 503 m, 461 w, 420 w.

### 2.2.5. [Tb$_2$Cd$_2$(tbtpy)$_2$(pfb)$_{10}$]×2MeCN (3$_{Tb}$)

The **3$_{Tb}$** was obtained according to the synthetic method similar to **2$_{Tb}$** using tbtpy (0.028 g, 0.098 mmol) instead tpy. The yield of **3$_{Tb}$** was 0.105 g (61%) based on [Tb$_2$Cd$_2$(MeCN)$_6$(pfb)$_{10}$]·4MeCN). Calculated for C$_{128}$H$_{76}$O$_{20}$N$_8$F$_{50}$Tb$_2$Cd$_2$ (%): C, 43.4; H, 1.7; N, 3.2. Found (%): C, 43.7; H, 1.4; N, 3.4. IR ($\nu$, cm$^{-1}$): 2971 w, 1932 w, 1735 m, 1679 m, 1618 s, 1592 s, 1490 s, 1299 w, 1250 m, 1202 w, 1108 s, 998 s, 934 w, 889 m, 829 m, 761 s, 740 s, 699 m, 610 m, 585 w, 510 w, 460 m, 425 w.

### 2.2.6. [Eu$_2$Cd$_2$(tppz)(pfb)$_{10}$]$_n$ (4)

Compound **4** was obtained like complex **2$_{Eu}$** using tppz (0.040 g, 0.108 mmol) instead tpy. The yield of **4** was 0.129 g (79.6% based [Eu$_2$Cd$_2$(MeCN)$_4$(pfb)$_{10}$]. Calculated for C$_{94}$H$_{16}$O$_{20}$N$_6$F$_{50}$Eu$_2$Cd$_2$ (%): C, 37.3; H, 0.5; N, 2.8. Found (%): C, 37.2; H, 0.4; N, 3.0. IR ($\nu$, cm$^{-1}$): 2988 w, 2945 w, 2901 w, 1651 m, 1627 m, 1606 s, 1526 m, 1495 s, 1437 m, 1412 m, 1375 s, 1297 m, 1158 m, 1106 m, 1038 s, 992 s, 942 s, 918 m, 895 w, 849 m, 749 s, 694 s, 637 m, 627 w, 560 s, 462 m.

### *2.3. X-ray Diffraction Studies*

Single crystal X-ray diffraction experiments were done on a Bruker Apex II diffractometer (Bruker, Billerica, MA, USA) with a CCD camera and a graphite monochromated

MoK$\alpha$ radiation source ($\lambda$ = 0.71073 Å) [29]. Semiempirical absorption corrections were applied for all the experiments using SADABS (University of Gottingen, Gottingen, Germany). [30]. Direct methods were used in structure solving. The refinement was done by the full-matrix least squares technique in anisotropic approximation for all non-hydrogen atoms. The H atoms were calculated geometrically and refined in the riding model. The calculations were performed in SHELX [31] using Olex2 (OlexSys Ltd, Chemistry Department, Durham University, DH1 3LE, UK) [32]. The SHAPE 2.1 software (University of Barcelona, Barcelona, Catalonia, Spain) [33] was used to determine the metals polyhedrons geometry.

The most important experimental crystallographic data and refinement statistics for 1, $2_{Eu}$, $3_{Eu}$, and 4 are reported in Table S1.

Powder X-ray diffraction data were collected using a Bruker D8 Advance diffractometer (Bruker, Billerica, MA, USA) (CuK$\alpha$, $\lambda$ = 1.54 Å, Ni-filter, LYNXEYE detector, geometry reflection).

### 2.4. Photo-Physical Measurements

The luminescence and excitation spectra were measured on a Horiba-Jobin-Yvon Fluorolog FL 3-22 (Horiba Scientific, Kyoto, Japan) spectrometer which has a 450 W xenon arc lamp as the excitation source for steady state measurements and 150W xenon pulse lamp for kinetic experiments. An R-928 PMT tube was used as a detector. Emission and excitation spectra were corrected for instrumental responses. Photoluminescence lifetimes $\tau^{obs}$ were obtained with the same instrument in time-resolved mode using a xenon flash lamp. Quantum yields $Q_{Ln}^L$ of lanthanide-centered luminescence were determined by an absolute method with an integration sphere (G8, GMP SA, Renens, Switzerland). The estimated error for quantum yields was $\pm$ 10%. All complexes studied were powdered before measurements. The calculation of the intrinsic quantum yield $Q_{Ln}^{Ln}$ of the europium emission has been performed by means of the Werts' formula [34]

$$Q_{Ln}^{Ln} = \tau^{obs} A_{01} n^3 \frac{I_{tot}}{I_{01}}$$

where $A_{01}$ = 14.65 s$^{-1}$ is the rate of magnetic dipole transition, $\frac{I_{tot}}{I_{01}}$ is the ratio of the total integrated emission transitions $^5D_0 \rightarrow {}^7F_J$ (J = 0–6) to the integrated intensity of the magnetic dipole transition $^5D_0 \rightarrow {}^7F_1$. The refractive index $n$ was taken as 1.5 for all cases.

## 3. Results

### 3.1. Synthesis of Complexes

Crystals of the mononuclear complex [Cd(tpy)(pfb)$_2$] (1, Figure 1) were obtained upon the interaction of cadmium pentafluorobenzoate [{Cd(H$_2$O)$_4$(pfb)}$_n$·$n$(pfb)] with 2,2′:6′,2″-terpyridine (tpy) in acetonitrile (Schemes 1 and S1). We have applied an approach based on the use of a stable polynuclear fragment that is not destroyed upon coordination to metal ions of $N$-donor ligands to synthesize heterometallic complexes with tridentate ligands [35,36]. New molecular complexes [Ln$_2$Cd$_2$(tpy)$_2$(pfb)$_{10}$] (Ln = Eu ($2_{Eu}$), Tb ($2_{Tb}$)), [Ln$_2$Cd$_2$(tbtpy)$_2$(pfb)$_{10}$] (Ln = Eu ($3_{Eu}$), Tb ($3_{Tb}$) and coordination polymer [Eu$_2$Cd$_2$(tppz)(pfb)$_{10}$]$_n$ (4) have been obtained in the reaction of [Eu$_2$Cd$_2$(MeCN)$_4$(pfb)$_{10}$] or [Tb$_2$Cd$_2$(MeCN)$_6$ (pfb)$_{10}$] 4MeCN with tpy (in the ratio Cd:tpy = 2:1), 4.4′,4″-tri-*tert*-butyl-2.2′:6′,2″-terpyridine (tbtpy) (in the ratio Cd:tbtpy = 1:1) or 2,3,5,6-tetra-(pyridin-2-yl)pyrazine (tppz) (in the ratio Cd:tppz = 1:1), respectively (Schemes 1 and S1). The use of a stoichiometric amount of tpy (in the ratio Cd:L = 1:1) led to the destruction of the heterometallic metal framework of the initial {Ln$_2$Cd$_2$} compounds and the formation of complex 1, whereas when using a stoichiometric amount of tbtpy, the heterometallic metal frame in $3_{Ln}$ is preserved (Schemes 1 and S1). In the overwhelming majority of cases such heterometallic compounds are resistant to the action of large excesses of $N$-donor ligands, even chelating ones [37–39]. However, during the synthesis of complexes with anions of pentafluorobenzoic acid we have already encountered the fact that an excess of pyridine,

2,4-lutidine, or isoquinoline led to the destruction of the heterometallic metal framework and crystallization of homometallic compounds from solution [27]. An additional factor contributing to the destruction of the heterometallic molecule is the noticeably lower solubility of 1 compared to $2_{Ln}$. Obtained compounds were characterized by single-crystal and powder X-ray diffraction, IR-spectroscopy, CHN-analysis. The structures of 1, $2_{Eu}$, $3_{Eu}$, 4 were determined by X-ray diffraction; isostructurality of $2_{Ln}$, $3_{Ln}$ (Ln = Eu, Tb) were confirmed by PXRD (Figures S1–S3). The purity of the obtained compounds was verified by CHN analyses and powder X-ray diffraction (complexes 1–3, Figures S1–S3).

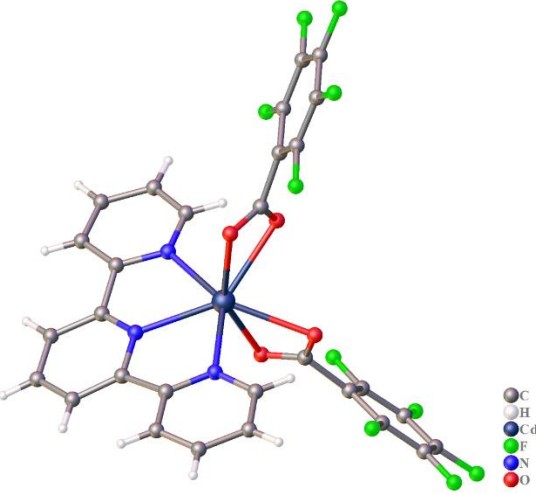

**Figure 1.** The molecular structure of 1.

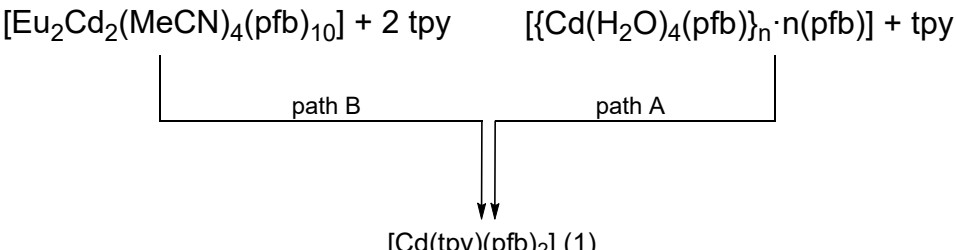

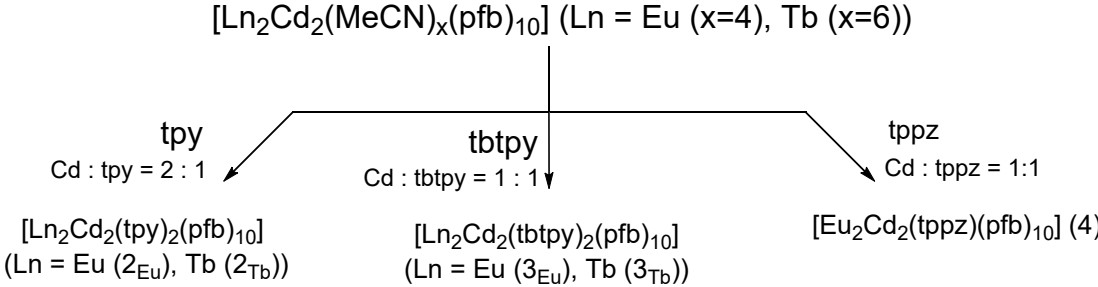

**Scheme 1.** Synthesis of complexes 1–4.

IR spectra of synthesized complexes 1–4 were analyzed. Carboxylic group vibrations of pentafluorobenzoate anions appear as peaks in 1589–1606 cm$^{-1}$ ($\nu$ asymmetric (COO$^{-}$)) and 1495–1488 cm$^{-1}$ ($\nu$ symmetric (COO$^{-}$)) wavelength range. C-F vibrations in the structures of pentafluorobenzoate anions lie in a wide range of 1390–1000 cm$^{-1}$ [40]. Bands in the range of 1520–1560 cm$^{-1}$ can be classified as C-C bond vibrations of pyridylic and phenylic aromatic fragments from pentafluorobenzoate anions and N-donor ligands.

*3.2. The Structure of Complexes*

Complex 1 (Figure 1) crystallizes in the orthorhombic space group *Pnna*. The co-ordination environment of cadmium ion is formed by four oxygen atoms (O(1), O(2), O(1A), O(2A)) of two chelate-bound pfb anions and three nitrogen atoms of the terpyridine molecule (N(1), N(1A), N(2)). The metal ion polyhedron is best described as pentagonal bipyramid (CdO$_4$N$_3$, Table S2). The coordinated tpy molecule has a nonplanar structure, and the angle between the plane of the central and side pyridyl rings in the ligand structure is 13.4(3)° and 9.0(3)°. In the crystal packing the molecules of the complexes are inter-connected by π-π, C-F...π (Table S3), C-H...O (Table S4) interactions with the formation of a supramolecular framework structure (the shortest distance between metal ions is 8.426(1) Å). The formation of π-π interactions is observed between pairs of tpy molecules of neighboring mononuclear fragments along the *b* axis (Figure S4), where the shortest distance between the centroids of aromatic fragments is 3.449(1) Å, and the angle between the planes is 4.44(13)°. The main bond lengths are presented in the Table S5.

For tpy the formation of mononuclear carboxylate complexes is very typical even in the case of metal ions that form rather large bond lengths and have large coordination numbers. Examples include Pb$^{II}$ acetate [41], Hg$^{II}$ trifluoroacetate [42], a fairly large number of carboxylate complexes of lanthanides and uranium [43–48]. Binuclear complexes in which a tpy molecule is coordinated to each metal atom can form in cationic carboxylate complexes with outer-sphere anions of strong acids, due to a decrease in steric hindrance [49–53]. The formation of binuclear complexes is also observed when small inorganic anions are combined with carboxylate anions [54–57].

Crystals of 2$_{Eu}$ and 3$_{Eu}$ are triclinic, space group *P$\bar{1}$*, (the inversion center is located between two atoms of Eu) they represent molecular complexes of very similar structure, in which the central europium atom are bonded to each other and to the terminal cadmium atom by bridging and chelate-bridging pfb-anions with the formation of a tetranuclear Cd-Eu-Eu-Cd metal core (Figure 2). Metal atoms lie in the same plane. The difference in the structure of the N-containing ligand leads to some changes in the structure of the complexes. Thus, while the cadmium coordination polyhedron in 2$_{Eu}$ is best described as a trigonal prism (Figure 3, Table S2, CdO$_3$N$_3$) formed by three bridged pfb anions and completed by three nitrogen atoms of the tpy molecule, the polyhedron of the corresponding atom in 3$_{Eu}$ represents a capped trigonal prism formed due to the fact that one of the pfb-anions acting as a bridge in 2$_{Eu}$ acts as a chelate-bridge in 3$_{Eu}$ forming an additional coordination bond with the cadmium atom. This results in the Cd-Eu distance in 2$_{Eu}$ of 4.2395(6) Å, which is slightly larger than that in 3$_{Eu}$ (4.209(2) Å). However, the Eu-Eu and Cd-Cd distance is slightly longer in 3$_{Eu}$ (3.8942(6) Å and 12.290(1) Å in 2$_{Eu}$ and 4.000(5) Å and 12.396(2) Å in 3$_{Eu}$).

Coordination environment of europium in compounds 2$_{Eu}$ and 3$_{Eu}$ consists of oxygen atoms of five bridging and two chelate-bridged pentafluorobenzoate anions (Figure 3) and is best described as a square antiprism in compound 2$_{Eu}$ and as a triangular dodecahedron in 3$_{Eu}$ (Table S2, EuO$_8$).

Molecules of coordinated tpy and substituted tpy in 2$_{Eu}$ and 3$_{Eu}$ structures have a non-planar structure and the angle between the plane of the central and lateral pyridyl rings in the structure of the ligand is 7.6(4)° and 11.1(4)° in 2$_{Eu}$ and 18.386(4)° and 18.839(1)° in 3$_{Eu}$.

In 2$_{Eu}$ and 3$_{Eu}$ the polyhedra of neighboring Eu ions have a common edge, (Figure 3). There are no common elements between the cadmium and europium polyhedra in 2$_{Eu}$, while in 3$_{Eu}$ there is a common vertex represented by the oxygen atom of the chelate-bridged pfb ligand. The values of the bond lengths M-O and Cd-N in the studied compounds are given in Table S5.

The crystal packing of 2$_{Eu}$ (Figure S5) is stabilized by C-H...F, C-H...O (Table S4), C-F...π (Table S3), F...F (Table S6) interactions with the formation of a supramolecular layer. The close-to-parallel orientation of tpy aromatic fragments of two adjacent molecules of the complex is also observed (the angle between the planes of aromatic fragments is 7.63°),

with a distance between the centroids of aromatic fragments of 3.876 Å, which may indicate the presence of weak $\pi \ldots \pi$ interactions.

A slight change in the structure of the compound $3_{Eu}$ does not lead to significant changes in the crystal packaging of the molecules (Figure S5). The main interactions that determine the structure of the crystal are also C-H...F, C-H...O, C-F...$\pi$ and F...F interactions (Tables S3, S4 and S6). The presence of three Me substituents in the tpy ligands leads to a weakening of the pi-pi interaction with their participation (the angle between the planes of aromatic fragments is 7.64°, a distance between the centroids of aromatic fragments is 3.937 Å), however, it does not have a great effect on the molecular packaging, which is the same in crystals of $2_{Eu}$ and $3_{Eu}$.

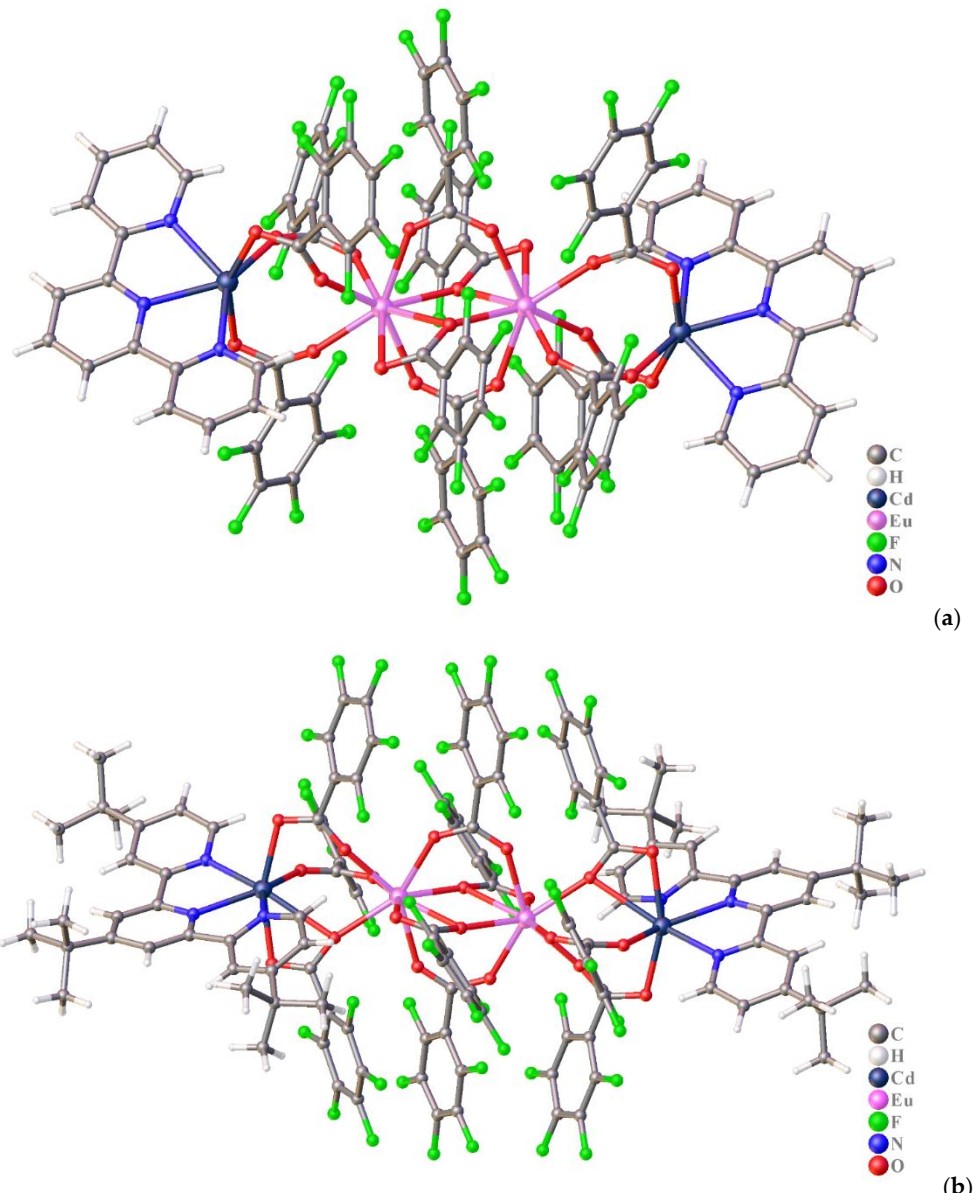

**Figure 2.** The molecular structure of $2_{Eu}$ (**a**) and $3_{Eu}$ (**b**).

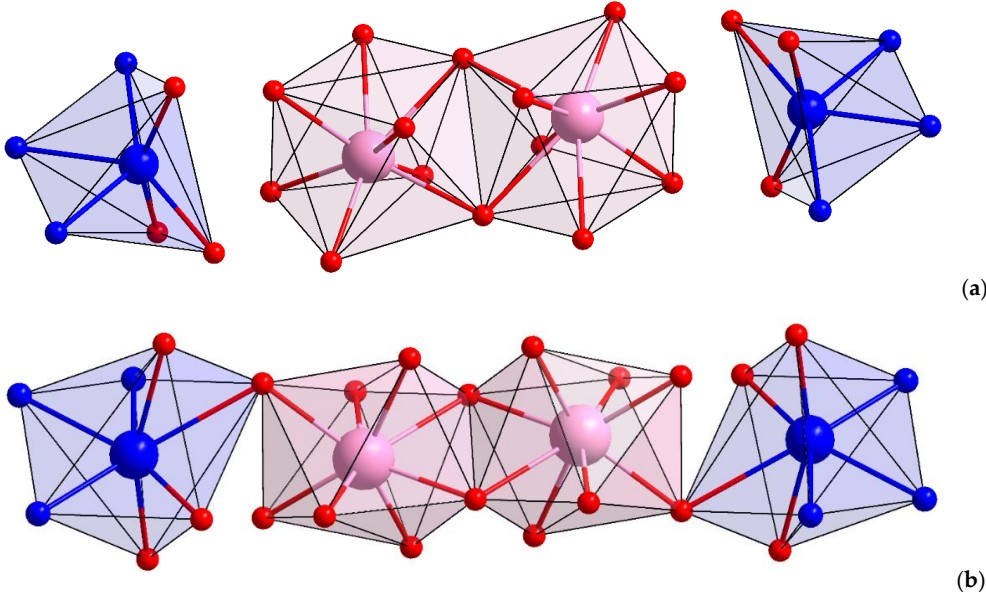

(a)

(b)

**Figure 3.** Coordination polyhedra of metal ions in 2$_{Eu}$ (**a**) and 3$_{Eu}$ (**b**).

To estimate the contribution of various non-covalent interactions to the crystal packages of 2$_{Eu}$ and 3$_{Eu}$ we have analyzed Hirschfeld surface using Crystal Explorer 17 program [58,59]. It was found that the largest contribution to the total Hirschfeld surface area of 2$_{Eu}$ is made by C-H...F (38.2%), F...F (23.9%), and C-F...$\pi$ (18.0%) interactions (Figures S6 and S7). While the contribution of C-H...O and $\pi$...$\pi$ interactions is only 3.0 and 2.8%, respectively. Replacing tpy molecule with tbtpy in the case of 3$_{Eu}$ leads to a significant reduction in the contribution of C-F...$\pi$ interactions and to an increase in the contribution of C-H...$\pi$ interactions to the Hirschfeld surface. The contribution of non-covalent interactions to the total Hirschfeld surface for the 3$_{Eu}$ complex is C-H...F (39.8%), F...F (15.8%), C-H...$\pi$ (10.5%), C-F...$\pi$ (6.2%), C-H...O (3.9%), $\pi$...$\pi$ (1.8%)) (Figures S8 and S9).

Complex 4 crystallizes in the triclinic space group $P\bar{1}$ (the inversion center is located between two atoms of Eu). Polymer chain in 4 consists of tetranuclear {Eu$_2$Cd$_2$(pfb)} fragments interconnected by 2,3,5,6-tetra-(pyridin-2-yl)pyrazine bridging molecules (Figure 4). In the tetranuclear {Eu$_2$Cd$_2$} fragment the Eu atom are bonded to each other and to Cd ions by bridging and chelate-bridging pfb-anions (Eu...Eu 3.906(1) Å, Cd...Cd 3.751(1) Å, Cd...Eu...Eu 166.4(1)°). The cadmium atom completes its environment to snub disphenoid by coordination of three nitrogen atoms of the bridging tppz molecule (Figure 5. Table S2, Cd O$_5$N$_3$), which, according to CCDC data, is a typical type of coordination of the tppz molecule to the cadmium ion. The coordination environment of Eu in 4 is best described as capped square antiprism (Table S2, EuO$_9$) consisting of oxygen atoms of five chelate-bridging and two bridging pfb-anions (Figure 5).

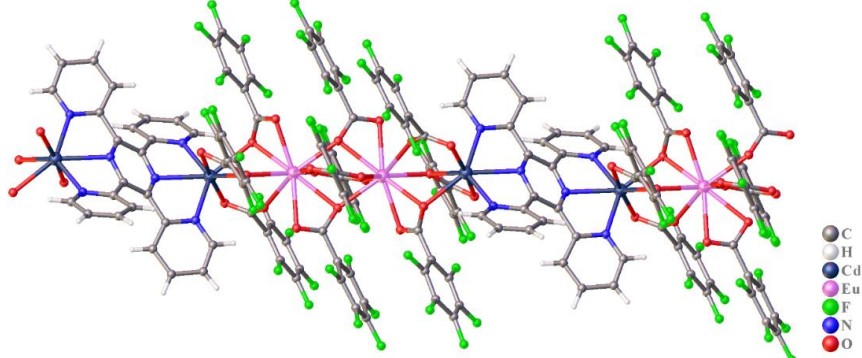

| | |
|---|---|
| ⬤ | C |
| ⬤ | H |
| ⬤ | Cd |
| ⬤ | Eu |
| ⬤ | F |
| ⬤ | N |
| ⬤ | O |

**Figure 4.** The polymer chain fragment of compound 4.

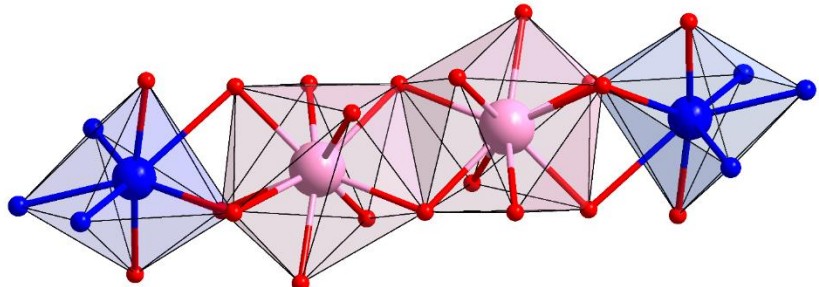

**Figure 5.** Coordination polyhedra of metal atoms in 4.

A crystal packing of 4 (Figure S10) is additionally stabilized via F...F (Table S6) and C-F...$\pi$ (Table S3) interactions with the formation of a supramolecular tridimensional structure.

Non-planar geometry typical for the coordinated tppz molecule [60–63] is also observed in the case of 4, where the pyridyl fragments are rotated relative to the plane of the central pyrazine fragment (Figure 4). The angle between the planes of the pyrazine and pyridyl fragments is 139.6(7)° and 143.4(6)°. In 4 the polyhedra of Eu and Cd atoms have a common edge (Figure 5).

Comparison of the geometrical parameters of the {$Eu_2Cd_2$} metal fragments in complexes $2_{Eu}$ and 4 showed that the replacement of the tpy molecule by tppz leads to a significant change in the geometry of the cadmium atom: an increase in the coordination number from six to eight due to additional coordination of oxygen atoms in the chelate-bridging carboxyl groups, as well as a decrease in the Cd ... Eu distance from 4.239(1) to 3.751(1) Å, whereas the geometry of the central fragment {$Eu_2$} and the value of the angle Cd-Eu-Eu do not undergo significant changes. The structure of the metal fragments of complexes $2_{Eu}$, $3_{Eu}$ and 4 also did not differ significantly from the previously obtained molecular pentafluorobenzoate complexes {$Ln_2Cd_2(pfb)_{10}$}, where cadmium atoms completed their environment due to the coordination of monodentate *N*-donor ligands [27,28]. A similar situation was also observed in the case of the previously obtained pivalate (piv) tetranuclear Cd-Li complexes [$Cd_2Li_2(L)_2(piv)_6$], in which the replacement of the monodentate or bidentate chelating ligand by tpy did not lead to significant distortion of the {$Cd_2Li_2$} metal framework [35], whereas in the case of using 1,10-phenanthroline (phen) as the N-donor ligand, a significant distortion of the tetranuclear metal backbones of pentafluorobenzoate complexes and the formation of polymer structures was observed [25]

### *3.3. Photoluminescence*

The photoluminescence properties of the solid-state $Eu^{3+}$ and $Tb^{3+}$ complexes $2_{Ln}$ and $3_{Ln}$ (Ln = Eu, Tb) are studied in detail. The compounds demonstrate bright luminescence due to *4f* transitions of lanthanide ions. The luminescence spectra of $2_{Eu}$ and $3_{Eu}$ demonstrate transitions $^5D_0$-$^7F_J$ (*J* = 0–4) of $Eu^{3+}$ (Figure 6). Weak bands are observed at 530–560 nm and correspond to transitions from higher excited state $^5D_1$ and indicates an incomplete intersystem conversion $^5D_1$-$^5D_0$. Similar splitting pattern of $2_{Eu}$ and $3_{Eu}$ refers to similar molecular structure of the compounds. The integration intensity ratios of $^5D_0$-$^7F_2$ and $^5D_0$-$^7F_1$ transitions of complexes $2_{Eu}$ and $3_{Eu}$ are 5.13 and 5.47, respectively. That indicates small deviation of $Eu^{3+}$ site symmetry from inversion center. Single symmetrical narrow component of $^5D_0$-$^7F_0$ transition in the luminescence spectra of $2_{Eu}$ and $3_{Eu}$ correlates with only one type of the $Eu^{3+}$ environment.

The narrow emission bands of $2_{Tb}$ and $3_{Tb}$ are assigned to $^5D_4$-$^7F_J$ (*J* = 6–0) transitions of $Tb^{3+}$ (Figure 7). The transitions $^5D_4$-$^7F_{2-0}$ demonstrate very low intensity while $D_4$-$^7F_5$ transition at 545 nm dominates in the emission spectra of $2_{Tb}$ and $3_{Tb}$ compounds and accounts for the green color of terbium luminescence.

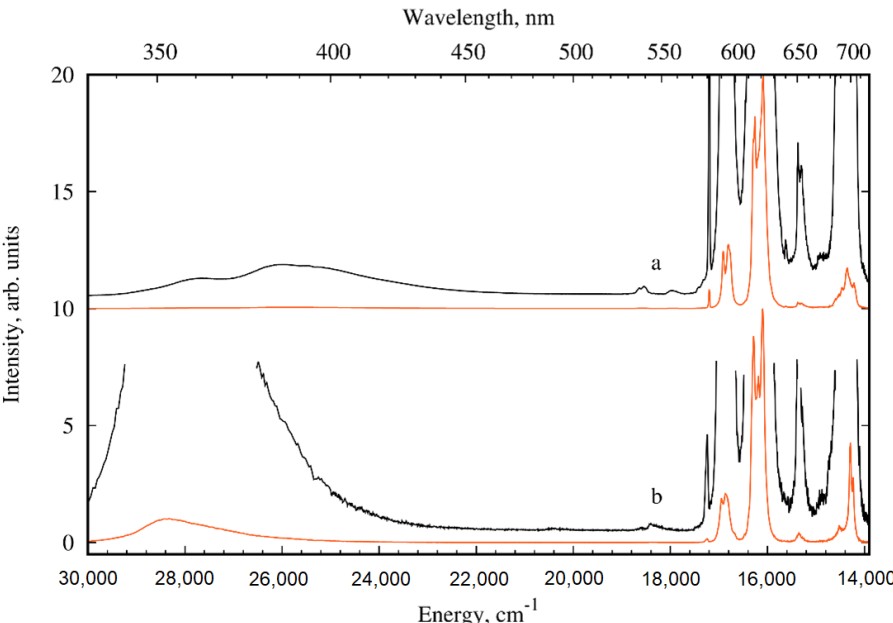

**Figure 6.** Luminescence spectra of $2_{Eu}$ (**a**) and $3_{Eu}$ (**b**) at $\lambda_{ex}$ = 280 nm and $T$ = 300 K in solid state. The black curves show the spectra after magnification.

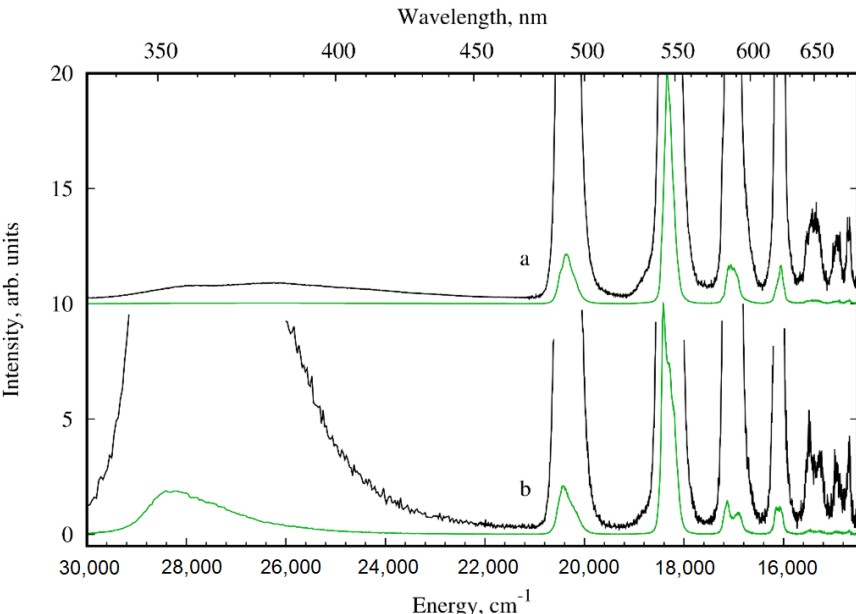

**Figure 7.** Luminescence spectra of $2_{Tb}$ (**a**) and $3_{Tb}$ (**b**) at $\lambda_{ex}$ = 280 nm and $T$ = 300 K in solid state. The black curves show the spectra after magnification.

The characteristic feature of the most heterometallic compounds with transition metals and lanthanide ions is the presence of the broad-band emission associated with *d*-block luminescence [25,64]. Investigated compounds demonstrate wide bands in the 330–400 nm wavelength range that can be associated with fluorescence of tpy and dttpy ligands (Figure S11). The maximum of dttpy fluorescence band is blue-shifted by 6 nm as a result of influence of electron-donating methyl group on the electronic density of the ligand. Additional band observed in the luminescence spectra of 1, $2_{Eu}$ and $2_{Tb}$ in 360–460 nm spectral range and overlaps with the fluorescence band. This band can be correlated with transition from the change transfer state typical for organocomplexes with transition metals [65]. Ligand luminescence leads to a decrease in the efficiency of lanthanide sensitization. As it can be seen from the full emission spectra (Figures 6 and 7), the

intensity of the d-block luminescent bands for the complexes $2_{Ln}$ is higher in comparison with the intensity of lanthanide luminescence, than for complexes $3_{Ln}$.

The excitation spectra of $2_{Ln}$ and $3_{Ln}$ are displayed in Figure 8 and demonstrate the wide ligand absorption bands as well as *4f* transitions of the lanthanide ions. The broad band with maxima at 345 nm can be assigned to $S_1$ state of ligands tpy and dttpy. The domination of the wide absorption bands in the spectra proves the effective luminescence sensitization via the organic ligands.

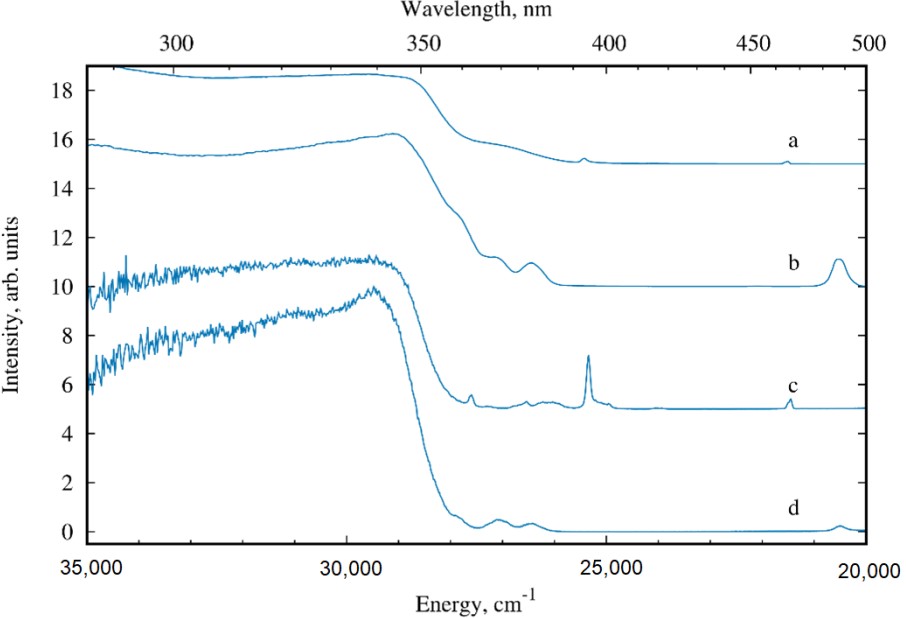

**Figure 8.** Excitation spectra of $2_{Eu}$ (**a**), $2_{Tb}$ (**b**), $3_{Eu}$ (**c**) and $3_{Tb}$ (**d**) at *T* = 300 K in solid state.

In addition to the characteristic excitation bands of lanthanide complexes, the spectra of $2_{Ln}$ compounds exhibit broad shoulders at 360–370 nm. Since this band is observed in both europium and terbium complexes, it cannot be assigned to a ligand to metal charge transfer state. The appearance of such bands was previously observed in complexes with 1,10-phenantroline [25,66] and 2,2'-bipyridine [67] due to π-stacking interaction. The stacking interaction leads to ligand charge redistribution and formation of intraligand charge transfer state. The weak bold shoulders in the spectra of complexes $2_{Ln}$ can be associated with charge transfer state as the result intermolecular interaction of tpy ligands. The shoulder is not observed in the excitation spectra of compounds $3_{Ln}$ as the result of weakening π-π interaction between dttpy by Me groups.

To estimate efficiency of energy transfer processes in the europium complexes, the photophysical parameters were calculated via Werts' formula [34] and presented in Table 1. Non-radiative rate constants values indicate only weak quenching processes of the lanthanide luminescence. However, compared with similar complexes $[EuCd(pfb)_5(phen)]_n$ and $[Eu_2Zn_2(pfb)_{10}(phen)_2]$ [25], $2_{Eu}$ and $3_{Eu}$ are characterized by more pronounced quenching of europium excited state which directly indicates the influence of the ligands tpy and dtppy. All investigated compounds demonstrate single exponential decay function of luminescence decay that confirm the lanthanide ion cite of one type. Very long lifetimes clearly reflect that lanthanide environment lacks an effective quencher as OH oscillators. Despite the presence of methyl groups in the ligand dttpy, the luminescence lifetimes and non-radiative rate constants of complexes $2_{Eu}$ and $3_{Eu}$ differ insignificantly due to the large distance of the C-H oscillators from the emission center and the rigid structure of the molecule, which is a unique feature for such heterometallic complexes.

**Table 1.** The radiative ($A_{rad}$) and non-radiative ($A_{nrad}$) decay rates, lifetimes ($\tau^{obs}$), intrinsic ($Q_{Ln}^{Ln}$) and overall ($Q_L^{Ln}$) quantum yields and sensitization effeciency ($\eta_{sens}$) of the complexes $2_{Eu}$, $2_{Tb}$, $3_{Eu}$ and $3_{Tb}$.

| Compound | $A_{rad}$, s$^{-1}$ | $A_{nrad}$, s$^{-1}$ | $\tau^{obs}$, ms | $Q_{Ln}^{Ln}$, % | $Q_L^{Ln}$, % | $\eta_{sens}$, % |
|---|---|---|---|---|---|---|
| $2_{Eu}$ | 370 | 300 | 1.49 | 55 | 39 | 71 |
| $2_{Tb}$ | - | - | 1.35 | - | 11 | - |
| $3_{Eu}$ | 410 | 380 | 1.27 | 52 | 31 | 60 |
| $3_{Tb}$ | - | - | 1.83 | - | 24 | - |

The energy of the lowest excited states of tpy equals to $E(S_1)$ = 29,000 cm$^{-1}$ and $E(T_1)$ = 22,000 cm$^{-1}$ [68]. The energy gap between $S_1$ and $T_1$ state of tpy equals to 7000 cm$^{-1}$ that significantly exceeds the optimal value 5000 cm$^{-1}$ [69]. The low energy of triplet state $T_1$ does not except the back energy transfer processes for $2_{Tb}$ compound as the energy gap between the $T_1$ and $^5D_4$ is just 1700 cm$^{-1}$. As a result the terbium compound demonstrates lower luminescence efficiency than europium analogue. It can be noted that the pentafluorobenzoic ligand can also participate in the process of energy transfer due to the rather low energy of triplet state ($E(T_1)$ = 22,300 cm$^{-1}$ [70]).

## 4. Conclusions

This work presents the result of the synthesis of new compounds based on the heterometallic tetranuclear metallocarboxylic core {Ln$_2$Cd$_2$(pfb)$_{10}$} (Ln = Eu, Tb) and tridentate tpy and tbtpy ligands, which are chelated to terminal cadmium atoms. It was shown that the {Eu$_2$Cd$_2$(pfb)$_{10}$} fragments with the tppz ligand form a 1D polymer due to the ligand's chelate-bridging function. These and earlier results show the stability of the {Ln$_2$Cd$_2$(pfb)$_{10}$} fragment with both mono-, bi-, and tridentate ligands. A series of $2_{Ln}$ and $3_{Ln}$ (Ln = Eu, Tb) exhibit metal-centered luminescence, but the presence of ligand luminescence bands indicates incomplete energy transfer from the *d*-block to the lanthanide ion, as well as a transition from the change transfer state in complexes with tpy ligand. Thus, the luminescence of europium and terbium ions in the tetranuclear fragment {Ln$_2$Cd$_2$(pfb)$_{10}$} is determined not only by coordinated pfb-anions, but also depends on the ligand in the coordination sphere of the terminal cadmium atom, which suggests the prospects for further design of brightly luminescent Ln$_2$Cd$_2$-complexes. The heterometallic compounds are effective phosphors due to the almost complete suppression of quenching processes and are promising for use as active materials in optoelectronic devices.

**Supplementary Materials:** The following are available online at https://www.mdpi.com/article/10.3390/inorganics10110194/s1, Figures S1–S3 (PXRD data), Figures S4–S10 (structural data), Figure S11 (photoluminescence data), Tables S1–S6, Scheme S1 (synthetic data).

**Author Contributions:** Conceptualization, validation, A.A.S. and I.L.E.; methodology, M.A.S. and A.A.S.; formal analysis, M.A.S., I.V.T., J.K.V. and F.M.D.; investigation, M.A.S., E.A.V. and M.A.E.; writing—original draft preparation, M.A.S. and A.A.S.; writing—review and editing, M.A.K., I.V.T. and J.K.V.; supervision, I.L.E. All authors have read and agreed to the published version of the manuscript.

**Funding:** This research was funded for Grants of the President of the Russian Federation, grant number MK-94.2022.1.3.

**Data Availability Statement:** Supplementary crystallographic data for the compounds synthesized are given in CCDC numbers 2133306 (1), 2133308 ($2_{Eu}$), 2207834 ($3_{Eu}$), 2133309 (4); These data can be obtained free of charge from The Cambridge Crystallographic Data Centre via www.ccdc.cam.ac.uk/data_request/cif.

**Acknowledgments:** X-ray diffraction analysis, CHN and IR-spectral analyzes were performed using the equipment at the Center for Collective Use of the Kurnakov Institute RAS, which operates with the support of the state assignment of the IGIC RAS in the field of fundamental scientific research. Photophysical measurements were carried out with the financial support from Ministry of Science and Higher Education of the Russian Federation using the equipment of Center for molecular composition studies of INEOS RAS.

**Conflicts of Interest:** The authors declare no conflict of interest.

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
