# Peer review of "Synthesis, Structure and Photoluminescence Properties of Cd and Cd-Ln Pentafluorobenzoates with 2,2′:6′,2′-Terpyridine Derivatives"

_inorganics, doi:10.3390/inorganics10110194_

Round 1

Reviewer 1 Report

Mikhail Kiskis and co-authors contribute to the emerging field of photofunctional materials describing the selective synthesis and properties of the series of d-/f–d mono- and polymetallic complexes supported by polypyridines and pentafluorobenzoate ligands. In addition to an extensive description of the XRD analysis accompanied by Hirschfeld surface analysis, the luminescent behavior of the polymetallic species has been briefly discussed. The crystallographic part of the provided manuscript is well-written and most of the statements are clearly supported, however, the emission behavior is missing some explanations and can be extended to prove the chosen concept. The general design of the using transition metal complex is hidden and can be emphasized across the manuscript, especially in the introduction and photophysical parts. Authors are also highly encouraged to reconsider extensive self-citation (17 papers out of 57, i.e. approx. 30%), and follow common good scientific practice (Nature 572, 578-579 (2019)). Nevertheless, the paper can be recommended for publishing after minor revisions are made. 

Several minor comments to the work addressed to the authors: 

-       IR-spectroscopy analysis of the species is not described, being the main characterization method apart from XRD. As only elemental analysis can be an indication of purity (I believe samples might suffer limited solubility in common organic solvents) it should be addressed in the paper as it prevents complete characterization. 

-       Figures 2 and 3 are missing colored agenda (colors of the fluorine atoms suggested to be exchanged as they are matching the color of Ln);

-       The general synthesis scheme is missing and could be added before the discussion part to ease the reading and improve the flow of the paper;

-       Shown B-alerts in the CIF-report files (structures 3 and 4) should be described and addressed;

-       Part of the photophysical description could be strengthened: a summary of the photophysical result can be considered; excitation spectra are likely to be replaced from ESI to the main text; full emission spectra can be represented as well.

-       The statement “The weak bold shoulders at 360-370 nm in the spectra of complexes 2Ln can be associated with charge transfer state as the result intermolecular interaction of tpy ligands.” is a bit speculative.

-       The emission behavior of the monometallic Cd complex is not considered and would be great to see a comparison with polymetallic ones. What is the actual influence of the secondary transition metal used in such systems? 

-       The emission spectra after magnification are not described and can be omitted.

-       Time-resolved data are expected to be attached to the manuscript and be discussed in more detail, other than giving descriptions such as “very long lifetimes”, and “lower luminescence lifetime”. The same applies to the calculation of the efficiency of energy transfer processes. 

Author Response

Thank you for your attention to the paper.

Comment

  1. I suggest authors to add a scheme detailing the main content of this study at the beginning of this manuscript.

Answer

Short and more detailed schemes have been added in the main text and supplementary materials, respectively.

Comment

  1. Can authors provide the PLQY of these complexes?

Answer

The PLQY were measured and included in the main text.

Comment

  1. INTRODUCTION section should be strengthened by emphasizing the novelty of this work.

Answer

The text of the introduction has been supplemented.

Reviewer 2 Report

This manuscript by Shmelev et al. reports the synthesis, structure and photoluminescence properties of Cd and Cd-Ln pentafluorobenzoates with 2,2′:6′,2'-terpyridine derivatives. This is a comprehensive study and the result seems good, I recommend a minor revision.

1. I suggest authors to add a scheme detailing the main content of this study at the beginning of this manuscript.

2. Can authors provide the PLQY of these complexes?

3. INTRODUCTION section should be strengthened by emphasizing the novelty of this work.

Author Response

Thank you for your attention to the paper. The self-citation percentage has been decreased to 20. Short and more detailed schemes have been added in the main text and supplementary materials, respectively. The photoluminescent part description was reworked significantly, quantum yields of synthesized complexes data were added and photoluminescence properties of the Cd compound were studied.

Comment

IR-spectroscopy analysis of the species is not described, being the main characterization method apart from XRD. As only elemental analysis can be an indication of purity (I believe samples might suffer limited solubility in common organic solvents) it should be addressed in the paper as it prevents complete characterization. 

Answer

A short description of IR spectra and CHN analysis data have been added.

Comment

-       Figures 2 and 3 are missing colored agenda (colors of the fluorine atoms suggested to be exchanged as they are matching the color of Ln);

Answer

The colours of REE ions in the corresponding pictures have been changed.

Comment

-       The general synthesis scheme is missing and could be added before the discussion part to ease the reading and improve the flow of the paper;

Answer

Short and more detailed schemes have been added in the main text and supplementary materials, respectively.

Comment

-       Shown B-alerts in the CIF-report files (structures 3 and 4) should be described and addressed;

Answer

Necessary clarifications have been added to the CIF files.

Comment

-       Part of the photophysical description could be strengthened: a summary of the photophysical result can be considered; excitation spectra are likely to be replaced from ESI to the main text; full emission spectra can be represented as well.

Answer

In accordance with these recommendations the description has been supplemented and the full spectra were presented in the main text.

Comment

-       The statement “The weak bold shoulders at 360-370 nm in the spectra of complexes 2Ln can be associated with charge transfer state as the result intermolecular interaction of tpy ligands.” is a bit speculative.

Answer

For clarity, this part has been rewritten in more detail with the addition of reference.

Comment

-       The emission behavior of the monometallic Cd complex is not considered and would be great to see a comparison with polymetallic ones. What is the actual influence of the secondary transition metal used in such systems? 

Answer

In accordance with the recommendation the emission spectrum was presented and the characteristic features of heterometallic complexes are described

Comment

-       The emission spectra after magnification are not described and can be omitted.

Answer

The magnification of the spectra makes it possible to see weak transitions of lanthanide ions, as well as weak luminescence bands of d-block.

Comment

-       Time-resolved data are expected to be attached to the manuscript and be discussed in more detail, other than giving descriptions such as “very long lifetimes”, and “lower luminescence lifetime”. The same applies to the calculation of the efficiency of energy transfer processes. 

Answer

The table with photophysical parameters has been added in the main text.